# Similarities and Differences in Extracellular Vesicle Profiles between Ischaemic Stroke and Myocardial Infarction

**DOI:** 10.3390/biomedicines9010008

**Published:** 2020-12-24

**Authors:** Laura Otero-Ortega, Elisa Alonso-López, María Pérez-Mato, Fernando Laso-García, Mari Carmen Gómez-de Frutos, Luke Diekhorst, María Laura García-Bermejo, Elisa Conde-Moreno, Blanca Fuentes, María Alonso de Leciñana, Eduardo Armada, Lorena Buiza-Palomino, Exuperio Díez-Tejedor, María Gutiérrez-Fernández

**Affiliations:** 1Neurological Sciences and Cerebrovascular Research Laboratory, Department of Neurology and Stroke Centre, Neuroscience Area of IdiPAZ, Hospital La Paz Institute for Health Research–IdiPAZ, La Paz University Hospital, Universidad Autónoma de Madrid, 28046 Madrid, Spain; oteroortega.l@gmail.com (L.O.-O.); elisaalonso164@hotmail.com (E.A.-L.); mery19832005@yahoo.es (M.P.-M.); fernilaso.9@gmail.com (F.L.-G.); mcarmen.gomezf@gmail.com (M.C.G.-d.F.); luke.diekhorst@gmail.com (L.D.); blanca.fuentes@salud.madrid.org (B.F.); malecinanacases@salud.madrid.org (M.A.d.L.); exuperio.diez@salud.madrid.org (E.D.-T.); 2Biomarkers and Therapeutic Targets Unit, Instituto Ramón y Cajal de Investigación Sanitaria (IRYCIS), 28034 Madrid, Spain; garciabermejo@gmail.com (M.L.G.-B.); elisa.condem@gmail.com (E.C.-M.); 3Acute Cardiac Care Unit, Cardiology Department, IdiPAZ Health Research Institute, 28046 Madrid, Spain; earmadarom@gmail.com; 4Clinical Analysis Department, La Paz University Hospital, 28046 Madrid, Spain; loren_337@hotmail.com

**Keywords:** exosomes, extracellular vesicles, ischaemia, miRNAs, myocardial infarction, proteins, stroke

## Abstract

Extracellular vesicles (EVs) are involved in intercellular signalling through the transfer of molecules during physiological and pathological conditions, such as ischaemic disease. EVs might therefore play a role in ischaemic stroke (IS) and myocardial infarction (MI). In the present study, we analysed the similarities and differences in the content of circulating EVs in patients with IS and MI. This prospective observational study enrolled 140 participants (81 patients with IS, 37 with MI and 22 healthy controls [HCs]). We analysed the protein and microRNA content from EVs using proteomics and reverse transcription quantitative real-time polymerase chain reaction and compared it between the groups. In the patients with IS and MI, we identified 14 common proteins. When comparing IS and MI, we found differences in the protein profiles (apolipoprotein B, alpha-2-macroglobulin, fibronectin). We also found lower levels of miR-340 and miR-424 and higher levels of miR-29b in the patients with IS and MI compared with the HCs. Lastly, we found higher miR-340 levels in IS than in MI. In conclusion, proteomic and miRNA analyses suggest a relationship between circulating EV content and the patient’s disease state. Although IS and MI affect different organs (brain and heart) with distinct histological characteristics, certain EV proteins and miRNAs appear to participate in both diseases, while others are present only in patients with IS.

## 1. Introduction

Stroke and myocardial infarction (MI) are the leading causes of death worldwide, according to the World Health Organization [1,2]. Ischaemic stroke (IS) and MI share common risk factors, such as hypertension, dyslipidaemia and diabetes [2], a number of aetiologies (such as atherosclerosis) [3], and the same pathogenesis: acute thrombotic arterial occlusion causing the ischaemic cell death of the tissue perfused by that artery [3]. Although various tissues are affected, IS and MI are both ischaemic processes that share certain characteristics.

Advances in cell biology have improved our understanding of the molecular processes associated with the ischaemic damage in IS and MI. However, the cellular events associated with ischaemia in specific organs are largely unknown. Intercellular communication appears to play an important role in regulating cell responses during ischaemia, with extracellular vesicles (EVs) being one of the most important mediators of intercellular signalling [4]. EVs are involved in the intercellular transfer of molecules (miRNAs and proteins) during IS [5,6,7,8,9,10] and MI [11,12,13,14], and a particular miRNA/protein signature in circulating EVs has been suggested. However, there is a lack of information regarding the similarities and differences in the expression pattern of these molecules in the two diseases. Unravelling the content of EVs in IS and MI would be a major step forward in defining the similar and specific pathological processes underlying cerebral and myocardial damage during acute ischaemia and in discovering new targets aimed at preventing and treating an ischaemic event. This study aimed to analyse the content (miRNA and protein) of circulating EVs in patients with IS and MI compared with healthy participants and to determine the similar and specific EV content profile according to the affected organ.

## 2. Materials and Methods

This was an observational and prospective study conducted at La Paz University Hospital (Madrid, Spain) between February 2017 and June 2019. The study recruited patients with IS from the Stroke Unit of the Department of Neurology whose inclusion criteria were an age of 18 years or older, less than 24 h from symptom onset and a pre-stroke modified Rankin Scale score of ≤1. The study also recruited patients with MI from the Department of Cardiology, whose inclusion criteria were 18 years or older, less than 72 h from MI onset (when the patient was haemodynamically stable). No sex-based or race/ethnicity-based differences were present.

The shared exclusion criteria were a previous IS (other than the index stroke in the IS group), cerebral haemorrhage, transient ischaemic attack, dementia or brain tumour, drug or alcohol dependence, any clinical condition that precluded diagnosis or follow-up and participation in a clinical trial.

All patients underwent the standard treatment for their disease, including intravenous thrombolysis or mechanical thrombectomy for the IS group and percutaneous coronary intervention for the MI group.

The study additionally recruited healthy controls (HCs) who stated that they had no known disease and voluntarily agreed to participate by signing the informed consent document.

The study collected demographic data, vascular risk factors and clinical data on admission. For the patients with IS, the study collected the baseline stroke severity (using the National Institutes of Health Stroke Scale, NIHSS) and the outcomes at 3 months by using the NIHSS score and the modified Rankin Scale (mRS). For the MI patients, the study also collected information on the recurrence of MI, IS and hospitalization at 3 months, as well as the treatment undergone for IS and MI. Informed consent was obtained from all HCs and patients (or their proxies) after a detailed explanation of the nature and purpose of the study. All data management was governed by the principles of Spanish Law 14/2007 of July 3 on biomedical research, ensuring the confidentiality of all personal data. The clinical study was approved by the Clinical Research Ethics Committee of La Paz University Hospital (reference PI-2562). The original data are available upon reasonable request.

### 2.1. Sample Collection

We collected 9 mL of peripheral blood in 3 serum-separator tubes from patients. Samples were centrifuged at 3000 g for 15 min at 4 °C. We aliquoted 4 mL of serum from each patient in 3 tubes (2 mL for EV isolation, 1 mL for miRNA extraction and 1 mL for proteomics from the EVs), which were stored at −80 °C until analysis.

#### 2.1.1. Extracellular Vesicle Isolation

We extracted EVs from 250 µL of serum with the ExoQuick ULTRA EV precipitation solution (System Biosciences, Palo Alto, CA, USA), as previously described [15]. This method employed an additional purification step. This approach ensures the full capture of EVs and allows for the removal of greater than 90% of “contaminating” proteins (albumins and immunoglobulins) following the enrichment. Following the manufacturer’s instructions, we added 67 µL of ExoQuick to 250 µL of serum, mixed the solution, and then let it rest for 30 min at 4 °C. We then centrifuged the ExoQuick/serum mixture at 3000 g for 10 min at 4 °C. After removing the supernatant, we suspended the pellet and added it to bipartite resin columns to filter out the albumin and immunoglobulins. We centrifuged the columns at 1000 g for 30 s to obtain purified EVs, which were kept at −80 °C until use.

#### 2.1.2. Extracellular Vesicle Characterisation

Nanoparticle tracking analysis (NTA) employs light scattering and Brownian motion properties to analyse EV size and quantity in a liquid suspension as previously described [16]. In the present study, we performed NTA using the NanoSight LM10 system (Malvern Panalytical, Malvern, UK). We analysed EVs diluted in 300 µL of Phosphate Buffered Saline (PBS) in a working concentration of 1 × 10^7^–1 × 10^9^ particles/mL and recorded three 60-s videos at a shutter speed of 30.00 ms and camera level of 13. For the analyses, we employed a detection threshold of 3, which was run in triplicate.

We employed transmission electron microscopy to visualise the EVs and verify that their size ranged from 30 to 150 nm as previously described [16]. The EVs were fixed in 2.5% glutaraldehyde 0.1 M sodium cacodylate solution for 1 h at 4 °C and postfixed with 2% osmium tetroxide for 1 h at 4 °C. We dehydrated the EV pellet with a graded acetone series and then embedded it in resin. We then cut 60-nM-thick sections and observed them under transmission electron microscopy at 80 kV, which was run in triplicate.

We employed western blot analysis to determine the presence or absence of EV proteins (ALIX, CD63 and TAPA-1), as previously described [16], and employed albumin as a negative control. We lysed the EVs with a radioimmunoprecipitation assay buffer (89900, Thermo Scientific, Waltham, MA, USA) and subjected them to western blot analysis using a 4–10% sodium dodecyl sulphate (SDS)-polyacrylamide gel (PAGE) for electrophoresis with 20 µg of protein per lane (quantified by bovine serum albumin). We employed the following antibodies: anti-CD63 (AB134045, Abcam, Cambridge, UK), anti-TAPA-1 (AB109201, Abcam, Cambridge, UK) and anti-Alix (2171, Cell Signal, Danvers, MA, USA). This procedure was run in triplicate.

### 2.2. Sample Preparation for Mass Spectrometry

We analysed the similarities and differences in EV proteomic composition among the groups by pooling the samples from patients of each group (IS, MI and HC). The identified proteins were compared among the groups.

Following the completion of the ExoQuick ULTRA protocol, the EVs were lysed in bulk, and the proteins were extracted from a PAGE gel. For the preparation of samples, we adjusted the buffer composition of the submitted samples to contain 2% SDS, 150 mM sodium chloride, and 50 mM TrispH8. We lysed the samples with a sonic probe (Q Sonica, Newtown, CT, USA) and heated them at 100 °C for 10 min. The protein concentration of the protein extracts was determined by the Qubit fluorometer. We processed 10 µg of each sample by SDS-PAGE using a 10% BisTrisNuPage mini-gel (Invitrogen, Waltham, MA, USA) in the 2-(N-morpholino) ethanesulfonic acid buffer system. We excised the migration windows (1-cm gel lane) and processed them by in-gel digestion with trypsin using a ProGest robot (DigiLab, Hopkinton, MA, USA) with the following protocol: (1) washing with 25-mM ammonium bicarbonate followed by acetonitrile, (2) reduction with 10-mM dithiothreitol at 60 °C, (3) alkylation with 50-mM iodoacetamide at room temperature, (4) digestion with trypsin (Promega, Madison, WI, USA) at 37 °C for 4 h, and (5) quenching with formic acid. We then directly analysed the supernatant without further processing.

#### Mass Spectrometry

We analysed half of the digested sample using nanoscale liquid chromatography coupled to tandem mass spectrometry (MS/MS) with a Waters^®^ nanoACQUITY HPLC system interfaced to a Q Exactive mass spectrometer (Thermo Fisher, Waltham, MA, USA). Peptides were loaded on a trapping column and eluted over a 75-μm analytical column at 350 nL/min. Both columns were packed with Luna C18 resin (Phenomenex, Torrance, CA, USA). The mass spectrometer was operated in a data-dependent mode, with the Orbitrap operating at 70,000 full width at half maximum (FWHM) and 17,500 FWHM for MS and MS/MS, respectively. We selected the 15 most abundant ions for MS/MS and employed 2 h of instrument time for the analysis of each digest.

We searched the processing data using a local copy of the software search engine Mascot (Matrix Science, London, UK) with the following parameters: Enzyme–Trypsin/P; Databases–Swiss Prot Human (concatenated forward and reverse plus common contaminants); Fixed modifications–Carbamidomethyl (C); Variable modifications–Acetyl (N-term), Deamidation (N, Q), Oxidation (M), Pyro-Glu (N-term Q); Mass values–Monoisotopic; Peptide Mass Tolerance–10 ppm; Fragment Mass Tolerance–0.02 Da; and Max Missed Cleavages–2.

Mascot DAT files were parsed into Scaffold (v.4.10.0m Proteome Software Inc., Portland, OR) for validation and filtering and to create a non-redundant list per sample. We filtered the data using a 1% protein and peptide false discovery rate (FDR) and requiring at least 2 unique peptides per protein. We also used Scaffold to validate the MS/MS-based peptide and protein identifications, accepting the peptide identifications if they could be established at >7.0% probability to achieve an FDR <1.0% by Scaffold’s local FDR algorithm. We accepted the protein identifications if they could be established at >99.0% probability to achieve an FDR <1.0% and contained at least 2 identified peptides. Protein probabilities were assigned by the Protein Prophet algorithm. Proteins that contained similar peptides and could not be differentiated based on the MS/MS analysis alone were grouped to satisfy the principles of parsimony.

### 2.3. RNA Extraction and Reverse Transcription Quantitative Real-Time Polymerase Chain Reaction

MiRNAs are small (18–28 nucleotides) noncoding RNAs that negatively modulate the expression of their target genes [17]. We analysed the similarities and differences in miRNA presence among the groups and analysed whole blood microRNAs enriched with those coming from EVs [18]. Prior to RNA isolation, we added a synthetic RNA (spike-in) to serum samples, which served as a technical control for extraction homogeneity by further spike-in amplification. We isolated total RNA enriched in miRNAs using the RNeasy mini kit (Qiagen, Germantown, MD, USA), starting with 200 µL of serum.

#### 2.3.1. cDNA Synthesis

To assess the cDNA synthesis efficiency, we added an external RNA (cell-miR-39) and further amplified it. For the cDNA synthesis, we employed the Universal RT miRNA PCR System (Qiagen Germantown, MD, USA). Briefly, 4 µL of RNA was employed as a template for reverse transcription in a final volume of 20 μL.

#### 2.3.2. MiRNA Array Profiling

Four patients per group were selected for the quantitative polymerase chain reaction (PCR) hybridization array screening, as previously described [19]. These four patients per group were chosen for homogeneous and similar lesions and matched in terms of age and sex. The hybridization array screening of 752 miRNAs was performed using the miRCURY locked nucleic acid (LNA) RT Kit (Qiagen Germantown, MD, USA) and miRCURY LNA miRNA miRNome PCR Panels I + II (YAHS-312 YG-8, Qiagen Germantown, MD, USA). Standardization of the array data was performed in a normalized manner using the mean expression of all miRNAs exhibiting a cycle threshold (CT) of ≤34. The miRNA profiling identified a subset of 7 top miRNAs (miRNAs with an absolute value of the log fold change >1 that were differentially expressed in the various samples).

#### 2.3.3. Validation of Selected miRNAs

The selected miRNAs were quantified by quantitative reverse transcription PCR (qRT-PCR) and analysed in 40 patients with IS, 20 patients with MI and 20 HCs. RNA degradation was checked in all samples. We diluted cDNA 1:11 with nuclease-free sterile water and used 4 μL of dilution as a template for PCR, performing the PCR detection with SYBR^®^ Green and specific LNA probes for each selected miRNA (Qiagen Germantown, MD, USA). We performed all reactions in triplicate using a Light Cycler 480 instrument (Roche) and calculated the CT values using a second derivative method (Light Cycler 480 Software 1.5, Roche, Basel, Switzerland). MiRNA expression values are presented as ΔCT, which were calculated as follows: ΔCT = miRNA CT-housekeeping CT. We employed NormFinder and BestKeeper software to determine the most stable housekeeping miNAs. Lastly, we employed the mean of miR-103a-3p CT and miR-30c-5p CT as the normalizer housekeeping CT.

### 2.4. Bioinformatic Analysis

Vesiclepedia is an open-access, regularly updated database of proteins, RNAs and lipids found in EVs. To analyse whether the proteins have been previously described in EVs, we compared the proteins identified in our experiments with the latest version of the protein record in Vesiclepedia (2019) [20] For a better interpretation of molecular mechanisms of identified proteins from EVs from patients in the IS, MI and HC groups, we performed gene ontology (GO) and pathway analyses using Kyoto Encyclopedia of Genes and Genomes (*KEGG*) [21] and EnrichR [22]. To confirm and complement these analyses, we then employed the Panther protein classification analysis [23]. These proteins were classified based on biological processes and molecular function to determine the protein classes and cellular processes involved in whole EV secretion. These functions were compared among the groups. We also analysed the gene ontology of the most enriched proteins found in common in the EVs from the HC, MI, and IS groups. We analysed the proteins that were identified in common in the MI and IS groups and the interaction between them to deepen our understanding of the characteristics of EVs in both diseases. We studied the differences between IS and MI in molecular function terms for the 10 most differentially enriched proteins.

We also analysed the miRNA predicted target using a bioinformatics tool, MirTarget, which was developed by analysing thousands of miRNA-target interactions from high-throughput sequencing experiments [24]. Moreover, we analysed the protein–protein interaction networks for the overlapping proteins of both ischaemic events using UniProt [25] and STRING [26].

### 2.5. Statistical Analysis

For the sample size calculation, the power analysis indicated that, with nonparametric testing, at least 20 patients needed to be included in each group to achieve a significance level of 5% (alpha) and a potency of 80% (1-beta).

We then analysed the data using the statistical IBM SPSS 23 program and represented the data using GraphPad Prism 8 software. Demographics and clinical data were tested for normality using the Kolmogorov–Smirnov test for groups with more than 30 degrees of freedom and the Shapiro–Wilk test for groups with less than 30 degrees of freedom. The study groups’ demographics, vascular risk factors, clinical data and treatments were compared using the Kruskal–Wallis test with a *post hoc* Mann–Whitney U test, as well as Fisher’s exact test, as appropriate. The results are expressed as mean ± standard deviation (SD). The proteomic data is shown as a descriptive analysis. The data from the MiRNAs were analysed using GenEx v.6. Normalisation was performed against the mean expression of the total miRNAs with CT ≤ 34. The results are expressed as mean ± standard deviation. We analysed the data using an analysis of variance with Bonferroni *post hoc* test. *p*-values < 0.05 were considered significant at a 95% confidence interval.

## 3. Results

### 3.1. Clinical Study

The study screened 172 participants: 22 in the HC group, 100 in the IS group and 50 in the MI group. After evaluating the inclusion and exclusion criteria, we excluded 19 and 13 patients in the IS and MI groups, respectively (Figure 1).

Ultimately, the study included 81 patients with IS, 37 patients with MI and 22 HCs (Figure 1A). Table 1 lists the participants’ demographic data, vascular risk factors and clinical data. The patients with IS were older than those with MI, and the patients with MI were more frequently men. The patients with MI more frequently had a smoking habit, whereas hypertension was more frequent among the patients with IS. There were no other significant differences between the IS and MI groups (Table 1). A total of 22 (27.2%) IS patients were treated with intravenous thrombolysis (IVT), three (3.7%) with mechanical thrombectomy (MT) and 13 (16%) with combined IVT and MT. All the patients in the MI group received percutaneous coronary intervention. At three months of follow-up, median score at the NIHSS in the IS group was 0 (IQR 1) and at the mRS 1 (IQR 2). One patient in the IS group was dead at 3 months, and none in the MI group. No ischemic recurrences at 3 months were found in any of the IS and MI groups. The median of the time of the sampling after infarction onset was 1 (IQR 0) in IS patients and 2 (IQR 1.5) in MI patients (*p* = 0.001). However, the median of the time of sampling after treatment was 1 day (IQR 0) in both groups (*p* = 0.335). The etiological subtype of the IS patients was six atherothrombotic, twenty-five cardioembolism, twenty-four small vessel disease, six uncommon cause and twenty unknown etiology.

### 3.2. Extracellular Vesicle Characterisation

Isolated EVs display a typical size of 30–150 nm, as seen by NTA. The EV-specific markers CD63, TAPA-1 and ALIX were present in the EVs isolated from serum and albumin as a negative marker (Figure 1B,C). The double membrane and circular shape of the EVs were observed by transmission electron microscopy (Figure 1D).

### 3.3. Bioinformatic Analysis

#### 3.3.1. Vesiclepedia

More than 76.96% of the proteins identified in our study were found in the latest version of the human protein record in Vesiclepedia (2019) (Figure 2A).

However, our study found 1 protein from patients with IS, 28 proteins from patients with MI and 37 proteins from HCs that are not registered in Vesiclepedia (Appendix A).

#### 3.3.2. In Silico Gene Ontology Analysis of the Total Proteins of Each Group

The identified proteins in the EV samples from the IS, MI and HC groups were classified based on biological processes and molecular functions. The biological process terms showed that IS, MI and HC groups had 8 of the 10 most enriched gene ontology terms in common. These functions in common are biological regulation, biogenesis, cellular process, immune system process, localisation, metabolic process, multiorgan process and response to stimulus. However, the cellular developmental process was highly enriched in the MI and IS groups compared with the HC group (Table 2).

A bioinformatic analysis with Panther protein classification showed that Mannose-binding protein C (MBL2) was the protein involved in this process.

Classification based on molecular function terms showed no differences between groups. EVs from the IS, MI, and HC groups were related to binding, catalytic activity, molecular function regulation, structural molecule activity, transcription regulator activity and transporter activity (Table 3).

#### 3.3.3. Similarities in Gene Ontology Terms in Ischaemic Stroke and Myocardial Infarction

We found 14 proteins that were present in the IS and MI groups but not in the HC group: apolipoprotein L1 (APOL1), apolipoprotein C (APOC), C-reactive protein (CRP), glyceraldehyde-3-phosphate dehydrogenase (GAPDH), complement C4 (C4), MBL2, loricrin (LOR), tubulin alpha-1B (TUBA1B), desmoglein-1 (DSG1), desmocollin-1 (DSC1), skin-specific protein 3 (XP3), hornerin (HRNR), leucine-rich alpha-2-glycoprotein 1 (LRG1) and coagulation factor IX (F9), most of which showed a strong interaction (Figure 2B).

#### 3.3.4. Differences in Gene Ontology Terms in the 10 Most Enriched Proteins among the Ischaemic Stroke, Myocardial Infarction, and Healthy Control Groups

Apolipoprotein B (APOB), alpha-2-macroglobulin (a2MG) and fibronectin (FN) were 2-fold enriched in IS compared with MI. All of these proteins are involved in biological regulation, immune system processes, and response to stimulus. Type II cytoskeletal keratin and von Willebrand factor were 2-fold enriched in IS compared with HC. We found no difference in the 10 most enriched proteins between MI and HC (Figure 2C).

### 3.4. MiRNA Analysis

In a miRNA microarray study, we studied 742 miRNAs in 4 patients from each group. Among these, we selected miR29b-3p, miR424-5p, miR369-3p, miR340-5p, miR199a-5p, miR376a-3p, and miR1537 for validation by individual qRT-PCR in 40 patients with IS, 20 patients with MI and 20 HCs (Table 4 and Table 5).

When comparing miRNAs between the groups, we found lower levels of miR-340 and miR-424 in the IS and MI groups compared with HCs, higher miR-29b levels in the IS and MI groups compared with HCs and lower miR-340 levels in the MI group compared with the IS group (Figure 3).

We also found lower levels of miR-199a and miR-639 in the HCs compared with the IS group. All of these miRNAs have been identified in serum EVs and published in Vesiclepedia.

Lastly, we analysed miRNAs predicted targets and protein-protein interactions by bioinformatic analysis. Our study determined that low-density lipoprotein receptor-related protein (LRP1), APOB and FN are predicted targets of miR-340. Moreover, protein–protein interaction analysis revealed that LRP1 also interacts with MBL2.

## 4. Discussion

Although IS and MI are two ischaemic processes that occur in two organs that differ in cell composition, we found similarities in the protein and miRNA content of circulating EVs. In the patients with IS and MI, we identified 14 proteins that were not in the HCs (APOL1, APOC, CRP, GAPDH, C4, MBL2, LOR, TUBA1B, DSG1, DSC1, XP3, HRNR, LRG1, and F9). We also found lower miR-340 and miR-424 levels and higher miR-29b levels in the patients with IS and MI compared with the HCs. These findings might show that circulating EVs participate in common processes in response to ischaemia in both the brain and the heart. However, when comparing the EV content in IS and MI, we also identified differences in the protein profile (APOB, a2MG, FN) and miR-340 expression between IS and MI, suggesting an organ-specific EV response to ischaemia.

EVs are an attractive source for providing information about the ischaemic process in IS and MI, can be synthesized and released from heart cells in the event of MI or from brain cells after IS (passing through the blood-brain barrier) and can be detected in peripheral blood [27]. Circulating EVs possess advantages in terms of abundance and accessibility. Through liquid biopsies, EVs help broaden our knowledge of ischaemic processes non-invasively and solve the issues of limited samples compared with tissue biopsy [28]. Moreover, EVs have the advantage of containing a plethora of markers and several information carriers such as proteins, microRNAs, mRNAs, lipids, and metabolites from their originating cells, acting as an “all-in-one complex biomarker” during ischaemia [29]. This content is well-protected within a lipid membrane, making EVs highly stable markers in the blood circulation. Given that they homogeneously circulate in the blood, EVs capture the entire heterogeneous information regarding the ischaemia process. Circulating EVs can be analysed quickly and effectively to monitor the ischaemic process and changes in real-time, which is essential in a clinical setting [29].

### 4.1. Shared Proteins and miRNAs

APOL1, APOC1, MBL2, miR-29b and miR-424 were present inside the EVs of the patients with IS and MI and were not found in the HCs. APOL1 and APOC1 are involved in lipid metabolism and the formation of most cholesterol esters in serum. Disorders in cholesterol synthesis regulation predispose patients to the development of atherosclerosis in IS and MI [30,31,32]. Atherosclerosis is the most frequent etiology of MI and is involved in up to 25% of IS cases. Atherosclerotic plaque rupture with superimposed in situ arterial thrombosis is the most likely mechanism underlying ischaemia in both processes [30,33,34,35]. This study is the first to report these proteins inside EVs in relation to the ischaemic response after IS and MI.

Our study found the proteins CRP and C4 in the EVs from the patients with IS and MI but not from the HCs. Along with C4, a key factor in the complement system, CRP is considered a potent activator of the complement system, the latter of which is activated after IS and MI and contributes to tissue injury after ischaemia [36]. In particular, individuals with higher CRP levels are at increased risk of mortality and morbidity after MI [37] and poor outcomes after IS [38]. One of the novel findings of our study is that we found CRP and C4 within EVs, suggesting that both EV-derived proteins can participate in the inflammatory process after IS and MI.

We found lower miR-29b levels in the EVs of the patients with IS and MI than in the HCs. Previous studies have demonstrated that miR-29b overexpression induces protection by negatively regulating p53-associated apoptosis [39], which could indicate that miR-29b is involved in alleviating ischaemia-induced apoptosis of damaged brain and heart tissue. This finding makes EV-derived miR-29b an interesting target in diminishing cell injury and promoting postischaemic tissue protection.

Our study found MBL2 to be present and miR-424 to be upregulated in the patients with MI and IS but not in the HCs. MBL2 is involved in cell morphogenesis (acquiring specialized structural and/or functional features) during cell differentiation in tissue repair. A previous study determined that miR-424 regulation promotes angiogenesis after ischaemia [40], which might indicate a specific role for EV-derived MBL2 and miR-424 in tissue healing and repair after ischaemia-induced injury, regardless of whether the injured tissue is part of the brain or heart. Future preclinical studies should analyse whether the administration of EVs loaded with overexpressed miR424 and MBL2 could enhance the protection and repair mechanism after IS and MI, thereby discovering new targets aimed at treating both ischaemic events.

We found the proteins GAPDH, TUBA1B, LOR, HRNR, DSG1, and DSC1 in the EVs of the patients with IS and MI but not in the HCs, proteins that have not been previously linked to the ischaemic process. Future studies should analyse the involvement of these proteins in ischaemia.

### 4.2. Specific Proteins and miRNAs

We found enriched APOB, a2MG, LRP1 and FN protein levels, along with varying miR-340 levels, in the patients with IS but not those with MI or the HCs. APOB, a2MG and FN have been previously associated with atherosclerosis progression in IS [41,42,43]. LRP1 not only plays a major role in protecting against atherosclerosis by reducing inflammation and removing apoptotic cells from lesions through phagocytic cells [44], but also interacts with MBL224. These findings could be of interest for identifying the different mechanisms between IS and MI.

APOB, FN, and LRP1 are predicted targets of miR-340 [24]. Another study confirmed that miR-340 levels are down-regulated in the acute phase of ischaemic stroke [45]. Future preclinical studies should analyse whether modified EVs loaded with a low expression of APOB and FL and an overexpression of LRP1 and MBL2 could reduce atherosclerosis formation.

This study brings us a step closer to understanding the mechanisms by which EVs act in both ischaemic events in order to identify new molecular targets, opening up new prevention and therapeutic strategies for both diseases. The present study can help discover common and specific molecules to prevent the recurrence of second ischaemic events. Given that both ischaemic diseases have an overlap in EV content in terms of proteins and miRNAs, one of the implications of our findings would be the future modification of EV content by reducing atherosclerosis or the inflammatory response.

### 4.3. Study Limitations

This study has a number of limitations, the first of which was the timing of the patient enrolment. The patients with IS were included within 24 h after stroke onset, while the patients with MI were included within 72 h after onset. However, the median time for the blood collection after treatment was similar for the two groups. The second limitation was the heterogeneity of the clinical characteristics of the patients with IS and MI and the differing treatment undergone, parameters that could influence the results of the study. The demographic variables in our study that were significantly different between the patients with IS and those with MI were age, dyslipidaemia, hypertension and smoking. Although hypertension and dyslipidaemia might affect EV content [46,47], a previous study reported that the EV content did not differ by age group or smoking status [48]. To our knowledge, there are no studies that have analysed whether the treatment of patients with IS and MI influences the circulating EV profile, but as this is an important concern, future studies should aim at deepening this knowledge. The third limitation was that the EV profile was determined at only one time point. The advantage of the liquid biopsy is that it can be tested multiple times to show the dynamics of the entire treatment process. It would be interesting to examine the EV profile of the patients with MI and IS after a set period (2–6 months), which might determine whether the EV profile is ischaemia related or whether the type of treatment and clinical characteristics could influence these patients’ EV profile. Futures studies should analyse the number and content of blood EVs at several time points to broaden the knowledge of the role of EVs in ischaemia. The fourth limitation is the isolation method employed for the EVs. Ultracentrifugation is often regarded as the gold standard; however, the technique has significant drawbacks, such as low reproducibility and low EV RNA and protein yield [49]. Precipitation-based isolation methods for EVs have much higher reproducibility and yield [50], although this comes at the cost of purity [51]. In this study, we used a post-isolation step, eliminating significant amounts of contaminants, such as albumin and immunoglobulins. However, we conducted no direct comparison between the ExoQuick-ULTRA method and ultracentrifugation in serum.

## 5. Conclusions

Proteomic and miRNA analyses suggest a relationship between circulating EV content from patients with MI and IS with risk factors and aetiology, as well as with post-ischaemic mechanisms of injury and repair. As for the similarities in the EV profile of IS and MI patients, we found that APOL1 and APOC1-derived EVs might be related to risk factors such as atherosclerosis. Moreover, EV-derived CRP and C4 might be involved in the post-ischaemic immune response and that miR-29b, MBL2 and miR-424 might be involved in the endogenous protection and repair mechanism. Although IS and MI affect different organs (brain and heart) with distinct histological characteristics, the proteins and miRNA mentioned above appear to participate equally in both diseases. However, we also noted differences in the EV profile between the IS and MI patients. We found that APOB, FN, MBL2, and miR-340 could be involved in atherosclerosis formation in IS but not in MI, suggesting different processes underlying ischaemia, depending on the affected organ.

## Figures and Tables

**Figure 1 biomedicines-09-00008-f001:**
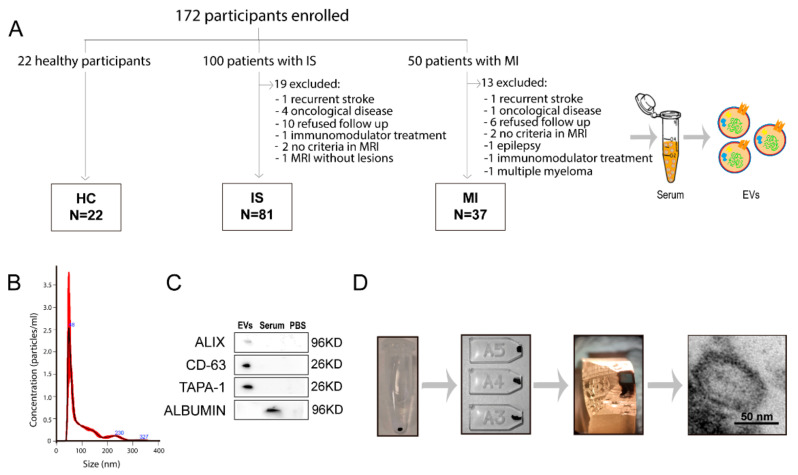
Collection and analysis of extracellular vesicles. (**A**) The number of patients from whom serum EVs were isolated. Characterization of EVs by size (**B**), EV phenotype by western blot (**C**) and their morphology by electron microscopy (**D**). Abbreviations: EVs, extracellular vesicles; HC, healthy controls; IS, ischaemic stroke; MI, myocardial infarction; MRI, magnetic resonance imaging; PBS, phosphate buffered saline.

**Figure 2 biomedicines-09-00008-f002:**
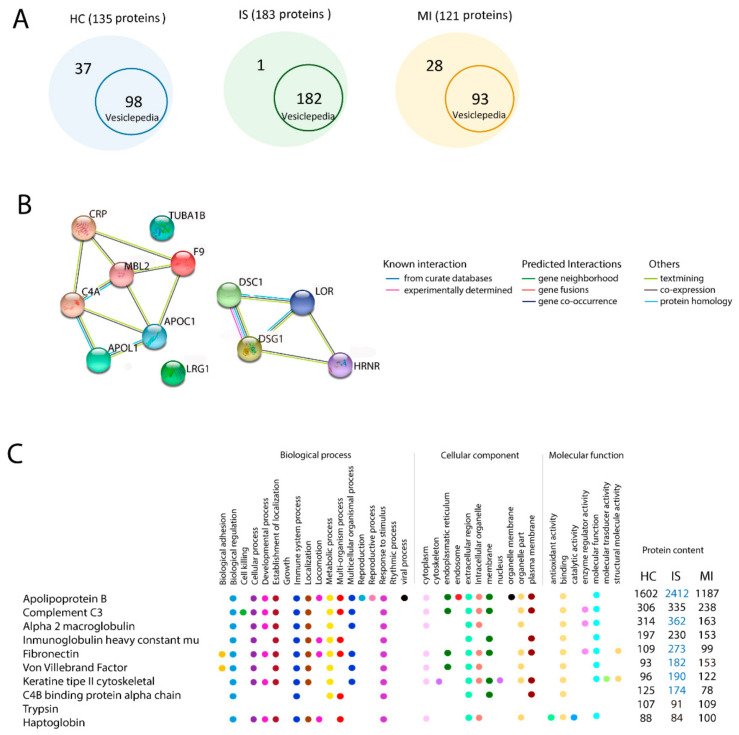
Proteomic analysis by mass spectrometry in extracellular vesicles. (**A**) Proteins identified in our experiments that have been previously found in the latest version of the human protein record in Vesiclepedia. (**B**) The 10 enriched proteins and their GO terms in the EV proteome from HC, IS, and MI. Comparison analysis for GO: biological process, cellular component and molecular function. Blue indicates whether the term is 2-fold more enriched in this group. (**C**) Interactions between the proteins identified in common for the IS and MI groups. Abbreviations: GO, gene ontology; HC, healthy controls; IS, ischaemic stroke; MI, myocardial infarction.

**Figure 3 biomedicines-09-00008-f003:**
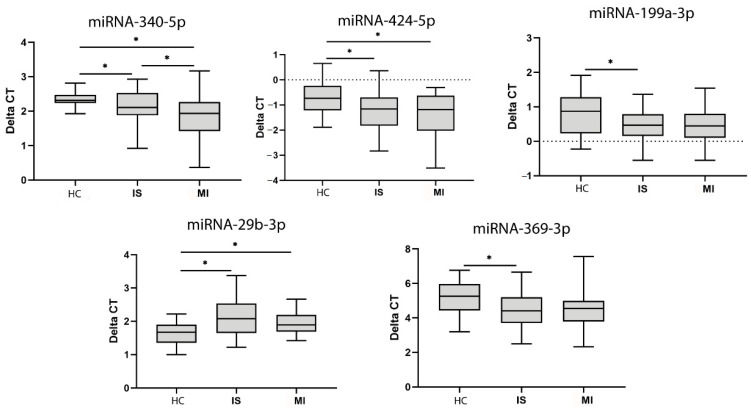
MiRNA expression in serum. MiRNAs with significant differences in expression identified after validation with qPCR. Abbreviations: HC, healthy controls; IS, ischaemic stroke; MI, myocardial infarction; CT, cycle threshold. Statistically significant values (*p* < 0.05) are indicated with an asterisk. * *p* < 0.05.

**Table 1 biomedicines-09-00008-t001:** Demographic data, vascular risk factors and clinical data of the patient cohort and healthy controls at admission.

Study Variables	HC *N* = 22	IS *N* = 81	MI *N* = 37	*p*
Mean age, years (SD)	61 (12.74)	67.675(14.88) *	55.95 (13.95)	**0.001**
Males, *N* (%)	7 (31.8)	42 (51.9)	27 (73) ^†^	**0.005**
Hypertension, *N* (%)	3 (13.63)	54 (69.2)*^,†^	17 (45.9)	**0.001**
Diabetes mellitus, *N* (%)	2 (9.09)	15 (19.2)	4 (10.8)	0.411
Dyslipidaemia, *N* (%)	3 (13.63)	36 (46.2) *^,†^	18 (48.6) ^†^	**0.007**
Smokers, *N* (%)	3 (13.63)	18 (22.2)*	22 (59.5) ^†^	**0.001**
Alcohol use, *N* (%)	1 (4.54)	6 (7.4)	1 (2.7)	0.761
Previous ischaemic Cardiomyopathy, *N* (%)	0 (0)	9 (11.8)	4 (10.8)	0.244
Median Charlson Comorbidity index (IQR)	0.00 (1)	1 (1.25) ^†^	1 (1) ^†^	**0.001**

Abbreviations: HC, healthy controls; IQR, interquartile range; IS, ischaemic stroke; MI, myocardial infarction; SD, standard deviation. Statistically significant values (*p* < 0.05) are in bold. * Mann–Whitney U test for continuous variables and Fisher’s exact test for categorical variables *p* < 0.016 to compare IS and MI groups. ^†^ Mann–Whitney U test for continuous variables and Fisher’s exact test for categorical variables *p* < 0.016 compared to HC.

**Table 2 biomedicines-09-00008-t002:** The enriched gene ontology biological processes in the healthy controls and patients with ischaemic stroke and myocardial infarction.

HC	IS	MI
Biological regulation	Biological regulation	Biological regulation
Biogenesis	Biogenesis	Biogenesis
Cellular process	Cellular process	Cellular process
Immune system process	**Cellular developmental process**	**Cellular developmental process**
Localization	Immune system process	Immune system process
Metabolic Process	Localization	Localization
Multiorgan process	Metabolic Process	Metabolic Process
Multicellular organismal process	Multiorgan process	Multiorgan process
Response to stimulus	Response to stimulus	Response to stimulus
Signalling	Signalling	Signalling

Abbreviations: HC, healthy controls; IS, ischaemic stroke; MI, myocardial infarction. Differences in biological processes are in bold.

**Table 3 biomedicines-09-00008-t003:** The enriched gene ontology molecular function in healthy controls and patients with ischaemic stroke and myocardial infarction.

HC	IS	MI
Binding	Binding	Binding
Catalytic activity	Catalytic activity	Catalytic activity
Molecular function regulation	Molecular function regulation	Molecular function regulation
Structural molecule activity	Structural molecule activity	Structural molecule activity
Transcription regulator activity	Transcription regulator activity	Transcription regulator activity
Transporter activity	Transporter activity	Transporter activity

Abbreviations: HC, healthy controls; IS, ischaemic stroke; MI, myocardial infarction.

**Table 4 biomedicines-09-00008-t004:** MiRNA candidates for validation.

**miRNAs HC vs. MI**	**Fold Change**	***p***
miR-376a-3p	2.804	**0.001**
miR-369-3p	4.055	**0.005**
miR-376c-3p	2.510	**0.006**
miR-150-5p	−2.485	**0.010**
miR-375	−3.007	**0.029**
miR-326	2.829	**0.033**
miR-197-3p	2.035	**0.035**
**miRNAs HC vs. IS**	**Fold Change**	***p***
miR-376a-3p	3.772	**0.001**
miR-376c-3p	3.353	**0.001**
miR-33a-5p	2.237	**0.001**
miR-100-5p	−2.181	**0.001**
miR-199a-5p	3.844	**0.001**
miR-369-3p	6.741	**0.001**
miR-326	4.287	**0.001**
miR-340-5p	2.304	**0.004**
miR-424-5p	2.588	**0.009**
miR-423-3p	2.000	**0.036**
miR-197-3p	2.320	**0.042**
miR-1537-3p	3.331	**0.046**
**miRNAs IS vs. MI**	**Fold Change**	***p***
miR-339-5p	−2.954	**0.001**
miR-181a-5p	−2.150	**0.001**
miR-194-5p	2.134	**0.001**
miR-100-5p	2.909	**0.004**
miR-29b-3p	−2.327	**0.005**
miR-15a-5p	2.004	**0.012**
miR-1537-3p	−3.148	**0.034**
miR-199a-5p	−2.168	**0.040**

MiRNAs found in the serum of patients with a fold change ≥2 and a *p*-value ≤0.05. Abbreviations: HC, healthy controls; IS, ischaemic stroke; MI, myocardial infarction.

**Table 5 biomedicines-09-00008-t005:** MiRNA validation.

miRNA	Patient Group		Patient Group	*p*
	MI		HC	
miR-424-5p	−1.331 ± 0.813	>	−0.773 ± 0.663	**0.023**
miR-340-5p	1.824 ± 0.7139	>	2.417 ± 0.372	**0.003**
miR-29b-3p	1.930 ± 0.3614	<	1.640 ± 0.362	**0.019**
	**IS**		**HC**	
miR-424-5p	−1.273 ± 0.747	<	−0.773 ± 0.663	**0.014**
miR-369-3p	4.510 ± 1.052	<	5.141 ± 0.976	**0.029**
miR-340-5p	2.168 ± 0.486	<	2.417 ± 0.372	**0.049**
miR-199a-3p	0.463 ± 0.451	<	0.796 ± 0.593	**0.019**
miR-29b-3p	2.093 ± 0.551	>	1.640 ± 0.362	**0.0001**
	**IS**		**MI**	
miR-340-5p	2.168 ± 0.486	>	1.824 ± 0.713	**0.03**

Significant values of miRNAs are expressed as the mean CT (cycle threshold) score ± SD. Abbreviations: HC, healthy controls; IS, ischaemic stroke; MI, myocardial infarction.

## Data Availability

The data presented in this study are available on request from the corresponding author.

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
