# Peer review of "Similarities and Differences in Extracellular Vesicle Profiles between Ischaemic Stroke and Myocardial Infarction"

_biomedicines, 2020, doi:10.3390/biomedicines9010008_

Round 1

Reviewer 1 Report

Laura and the co-authors have presented a very interesting study of “Organ-specific extracellular vesicles profile in acute ischemia. Similarities and differences between ischemic stroke and myocardial infarction”. In this study, they compared the content of circulating extracellular vesicles (EVs) in two ischemic processes and find similarities. As an important topic of liquid biopsy, EVs have a huge potential to be an effective tool for tumor diagnosis. Proteomics analysis of plasma-derived EVs can help to discover novel EV based biomarkers. However, some major points still need to be considered to further improve in this manuscript.

  1. For the title of the manuscript “Organ-specific extracellular vesicles profile in acute ischemia. Similarities and differences between ischemic stroke and myocardial infarction”, it might be better to combine two-sentence into one to make the topic more concise and clear.
  2. Figure 1C, EV phenotype by western blot, line 250.

Three typical positive protein markers (ALIX, CD63, TAPA1) have been detected in the research. However, it is suggested at least one negative protein marker for EVs phenotype according to the Minimal information for studies of extracellular vesicles 2018 (MISEV2018).

  1. The format of all the tables in the manuscript is not in a common style, and it is suggested to optimize into a three-line chart.
  2. Table 1: Demographic data of the patient cohort and health controls at admission, line 260.

Some basic information is documented in the table, and it is recommended to supplement the patient's follow-up treatment and prognosis information. These are also important to explore the connection between markers in EVs and patients' prognosis.

  1. One of the important advantages of liquid biopsy is that markers can be tested at multiple time points to show the dynamics in the whole treatment process. Is it possible to add the variations of EVs markers from the same patient during the treatment and prognosis, or may discuss as a future exploration goal?
  2. Concerning of the micro-RNAs as biomarkers, there are also a number of studies focus on the circulating miRNAs in the blood samples, or in the saliva samples. It might be interesting to compare or check the literature for more comprehensive evidence to indicate the benefit of circulating EV markers. And the combination of proteomic screened markers together with the microRNAs will be more comprehensive for the significance of the study

Author Response

Comments and Suggestions for Authors

Laura and the co-authors have presented a very interesting study of “Organ-specific extracellular vesicles profile in acute ischemia. Similarities and differences between ischemic stroke and myocardial infarction”. In this study, they compared the content of circulating extracellular vesicles (EVs) in two ischemic processes and find similarities. As an important topic of liquid biopsy, EVs have a huge potential to be an effective tool for tumor diagnosis. Proteomics analysis of plasma-derived EVs can help to discover novel EV based biomarkers. However, some major points still need to be considered to further improve in this manuscript.

  1. For the title of the manuscript “Organ-specific extracellular vesicles profile in acute ischemia. Similarities and differences between ischemic stroke and myocardial infarction”, it might be better to combine two-sentence into one to make the topic more concise and clear.

We appreciate your comments and suggestions as they provide us with the opportunity to improve our manuscript. Following your recommendations, we have combined the two-sentence title into one, which now reads as follows:  Similarities and differences in extracellular vesicle profiles between ischaemic stroke and myocardial infarction

  1. Figure 1C, EV phenotype by western blot, line 250.

Three typical positive protein markers (ALIX, CD63, TAPA1) have been detected in the research. However, it is suggested at least one negative protein marker for EVs phenotype according to the Minimal information for studies of extracellular vesicles 2018 (MISEV2018).

Following your recommendations, we have introduced albumin as a negative protein marker for EVs in line 136 of “Extracellular vesicle characterization” section and in the Figure 1.

  1. The format of all the tables in the manuscript is not in a common style, and it is suggested to optimize into a three-line chart.

Following your recommendations, we have changed the format of all tables (lines 285, 322, 332, 356 and 359).

  1. Table 1: Demographic data of the patient cohort and health controls at admission, line 260.

Some basic information is documented in the table, and it is recommended to supplement the patient's follow-up treatment and prognosis information. These are also important to explore the connection between markers in EVs and patients' prognosis.

Following your recommendations, we have included the patients’ follow-up treatment and prognosis information in lines 274-279. We have included this information in the text instead of Table 1 in order to make the table 1 clearer.

Thank you for your suggestion. It was fascinating to explore the connection between markers in EVs and the patients’ prognosis. In fact, we took this analysis into account in the present study. However, in the MI group, we found only 36 patients with good recovery and only 1 patient with a poor recovery. With this patient distribution, it is therefore not possible to perform a logistic regression or an ROC curve.

  1. One of the important advantages of liquid biopsy is that markers can be tested at multiple time points to show the dynamics in the whole treatment process. Is it possible to add the variations of EVs markers from the same patient during the treatment and prognosis, or may discuss as a future exploration goal?

This is undoubtedly a crucial aspect that we will take into account for future studies. In this case, this is a limitation that we have incorporated in a new section called Study Limitations (lines 464-471). The third point of this section reads, “The third limitation was that the EV profile was determined at only one time point. The advantage of the liquid biopsy is that it can be tested multiple times to show the dynamics of the entire treatment process. It would be interesting to examine the EV profile of the patients with MI and IS patients after a set period (2–6 months), which might determine whether the EV profile is ischaemia related or whether the type of treatment and clinical characteristics could influence these patients’ EV profile. Futures studies should analyse the number and content of blood EVs at several time points to broaden the knowledge of the role of EVs in ischaemia.”

  1. Concerning of the micro-RNAs as biomarkers, there are also a number of studies focus on the circulating miRNAs in the blood samples, or in the saliva samples. It might be interesting to compare or check the literature for more comprehensive evidence to indicate the benefit of circulating EV markers. And the combination of proteomic screened markers together with the microRNAs will be more comprehensive for the significance of the study

Following your recommendations, we have introduced new information in the Discussion section (lines 387-399), which now reads as follows: “EVs are an attractive source for providing information about the ischaemic process in IS and MI, can be synthesised and released from heart cells in the event of MI or from brain cells after IS (passing through the blood-brain barrier) and can be detected in peripheral blood27. Circulating EVs possess advantages in terms of abundance and accessibility. Through liquid biopsies, EVs help broaden our knowledge of ischaemic processes non-invasively and solve the issues of limited samples compared with tissue biopsy28. Moreover, EVs have the advantage of containing a plethora of markers and several information carriers such as proteins, microRNAs, mRNAs, lipids and metabolites from their originating cells, acting as an “all-in-one complex biomarker” during ischaemia29. This content is well-protected within a lipid membrane, making EVs highly stable markers in the blood circulation. Given that they homogeneously circulate in the blood, EVs capture the entire heterogeneous information regarding the ischemia process. Circulating EVs can be analysed quickly and effectively to monitor the ischaemic process and changes in real-time, which is essential in a clinical setting.29

We also agree with the reviewer regarding the studies profiling miRNA in blood and saliva that have already been defined as biomarkers in several diseases, including IS and MI. If statistical correlation between miRNA expression and disease features could be established, miRNAs could be considered useful biomarkers. miRNAs and other molecules contained in EVs have additional value as highly relevant molecules for the pathophysiological mechanism of diseases, given that EVs “per se” are signalling structures, responsible for systemic abnormalities in diseases. Therefore, miRNAs identified inside EVs could have double significance: as a biomarker and as part of the pathophysiological mechanisms responsible for the disease onset and progression.

Regarding the integration of proteomic and transcriptomic data, we agree with the reviewer that it would be highly interesting, given that it will provide an overall idea of the functional burden of exosomes. It is conceivable that some of the identified miRNAs could regulate the proteins that were also validated in the two cohorts. If there is no potential regulation between miRNAs and proteins, at least miRNAs and proteins could cooperate towards specific cell functions or processes for the signaling role of EVs in IS and MI as part of the pathophysiological mechanisms underlying the two diseases.

Reviewer 2 Report

Organ-specific extracellular vesicles profile in acute 2 ischemia. Similarities and differences between 3 ischemic stroke and myocardial infarction.

Organ-specific extracellular vesicles profile in acute 2 ischemia. Similarities and differences between 3 ischemic stroke and myocardial infarction.     Otero-Oretega/Alonso-Lopez et al. examined circulating EV cargoes (proteins and miRNAs) using high throughput proteomic and microarray approach. The study's strength was that it was a human study that could have a direct clinical implication. However, the review was inadequate in a few areas (see below). Moreover, the lack of findings and pathologic correlation makes the study too descriptive.   Major concerns: 1. Study samples: The main concern was the timing of sample collection. The window period of 24 h after IS onset and 72 h after MI onset add a significant variable to the data. It is challenging to interpret pathophysiology or clinical comparison if both groups likely received ischemia treatment, which would alter circulating EVs. It would be helpful to identify those variables in Table 1. These include the hour of blood collection after the onset of an ischemic event and the type of medical therapy. The difference between IS and MI patients' characteristics added some difficulty understanding the meaning of those EV cargoes; please further elaborate on this topic.   2. EV isolation method: precipitation method raises significant concern for impurity. Moreover, in Figure 1-C, instead of using PBS as a control lane, should the comparison be the EV-free plasma or serum? The TEM image was tough to discern EVs; please also add a scale bar.   3. miRNA profiling approach: it seems that miRNA sequencing may be more appropriate than miR-array study. The integration of proteomic and transcriptomic data may help examine the further clinical correlation between the differential expressions between IS and MI, as well as the meaning of comparing these 2 cohorts.   4. The results' overall interpretation was unfocused and unclear of the authors' aims for this study. Moreover, can the author add one conclusion of the results in the abstract?   Minor: GADPH should be read as GAPDH?

Author Response

Comments and Suggestions for Authors

Organ-specific extracellular vesicles profile in acute ischemia. Similarities and differences between ischemic stroke and myocardial infarction.

Organ-specific extracellular vesicles profile in acute ischemia. Similarities and differences between ischemic stroke and myocardial infarction.    

Otero-Ortega/ Alonso-Lopez et al. examined circulating EV cargoes (proteins and miRNAs) using high throughput proteomic and microarray approach. The study's strength was that it was a human study that could have a direct clinical implication. However, the review was inadequate in a few areas (see below). Moreover, the lack of findings and pathologic correlation makes the study too descriptive.   Major concerns: 

  1. Study samples: The main concern was the timing of sample collection. The window period of 24 h after IS onset and 72 h after MI onset add a significant variable to the data. It is challenging to interpret pathophysiology or clinical comparison if both groups likely received ischemia treatment, which would alter circulating EVs. It would be helpful to identify those variables in Table 1. These include the hour of blood collection after the onset of an ischemic event and the type of medical therapy. The difference between IS and MI patients' characteristics added some difficulty understanding the meaning of those EV cargoes; please further elaborate on this topic.

We appreciate your comments and suggestions as they provide us with the opportunity to improve our manuscript. Following your recommendations, a native English speaker has reviewed our manuscript.

In addition, we have included information on the timing of the blood collection after the onset of the ischaemic event and the type of medical therapy (lines 280-282). We have also incorporated a new section called Study Limitations to comment on this topic (lines 453-456), which now reads as follows: “This study has a number of limitations, the first of which was the timing of the patient enrolment. The patients with IS were included within 24 h after stroke onset, while the patients with MI were included within 72 h after onset. However, the median time for the blood collection after treatment was similar for the two groups.”

  1. EV isolation method: precipitation method raises significant concern for impurity. Moreover, in Figure 1-C, instead of using PBS as a control lane, should the comparison be the EV-free plasma or serum? The TEM image was tough to discern EVs; please also add a scale bar.  

The precipitation-based isolation of EVs is indeed known for containing more impurities than ultracentrifugation. However, ultracentrifugation also has several significant drawbacks including low reproducibility and low EV RNA yield. The isolation-based precipitation method employed additional purification, which removes the majority of albumin and immunoglobulins, leading to comparable or even cleaner results than those of ultracentrifugation. However, following your recommendations, we have included a new section called Study Limitations to comment on the EV isolation method (lines 471-478), which now reads as follows: “The fourth limitation is the isolation method employed for the EVs. Ultracentrifugation is often regarded as the gold standard; however, the technique has significant drawbacks, such as low reproducibility and low EV RNA and protein yield.49 Precipitation-based isolation methods for EVs have much higher reproducibility and yield,50 although this comes at the cost of purity.51 In this study, we used a post-isolation step, eliminating significant amounts of contaminants, such as albumin and immunoglobulins. However, we conducted no direct comparison between the Exoquick-ULTRA method and ultracentrifugation in serum.”

Regarding the western blot and transmission electron microscopy image, we have included EV-free serum as a control lane in Figure 1-C. We have also added a scale bar to the TEM image.

  1. miRNA profiling approach: it seems that miRNA sequencing may be more appropriate than miR-array study. The integration of proteomic and transcriptomic data may help examine the further clinical correlation between the differential expressions between IS and MI, as well as the meaning of comparing these 2 cohorts.  

We agree with the reviewer that deep sequencing is the most appropriate method for miRNA profiling in a “fishing” experimental approach. In our experience with other projects, however, sequencing approaches result in numerous sequences, some of which are unknown and are therefore new miRNAs. Given that the miRNAs identified after sequencing should be validated by qRT-PCR in larger cohorts, there are often no validated tools available for the amplification of unknown miRNAs. Therefore, for the present manuscript and taking into account that miRNAs are not the only molecules studied, we considered that qRT-PCR miRNA arrays testing up to 762 miRNAs from the already known 2800 would provide sufficient information and, more importantly, robust information, given that probes for the further validation of the selected miRNAs by qRT-PCR are assured. In fact, we obtained reproducible and robust results in miRNA expression with this workflow, as has been included in the manuscript.

Regarding the integration of proteomic and transcriptomic data, we agree with the reviewer that it would be interesting, given that it will provide an overall idea of the functional burden of exosomes. It is conceivable that some of the identified miRNAs could regulate the proteins that were also validated in the two cohorts. If there is no potential regulation between miRNAs and proteins, at least miRNAs and proteins could cooperate towards specific cell functions or processes for the signalling role of EVs in IS and MI as part of the pathophysiological mechanisms underlying the two diseases.

  1. The results' overall interpretation was unfocused and unclear of the authors' aims for this study. Moreover, can the author add one conclusion of the results in the abstract?   Minor: GADPH should be read as GAPDH?

Following your recommendations, we have rewritten the Discussion section to clarify the overall interpretation of the results, which is now more in line with the aim of the study (lines 400-451). We have also included a conclusion in the Abstract (lines 44-48).

Minor change: We have changed GADPH to GAPDH.
